# Expectations versus Reality of Designer Dog Ownership in the United States

**DOI:** 10.3390/ani12233247

**Published:** 2022-11-23

**Authors:** Bridget Hladky-Krage, Christy L. Hoffman

**Affiliations:** Department of Animal Behavior, Ecology, and Conservation, Canisius College, Buffalo, NY 14208, USA

**Keywords:** canine welfare, anthrozoology, designer dogs, dog breeds, dog ownership

## Abstract

**Simple Summary:**

“Designer dogs”, or the hybrid offspring of two purebred dogs, are an extremely popular pet choice in the United States. However, there are many misconceptions surrounding them, and the reality of owning one may not match the owner’s expectations. For instance, many people believe these dogs to be non-shedding, hypoallergenic, and low maintenance; however, this is not always the case. This study compared owner expectations and reality associated with owning purebred dogs, mixed-breed dogs, and designer breeds—specifically, doodles (i.e., poodle hybrids, such as Labradoodles and Cockapoos). We found that the decision to acquire a doodle was driven largely by the appearance of doodles and by the perception that they are good with children and generally healthy. Our data also showed that meeting the maintenance and grooming needs of doodle dogs required more investment than owners expected. This finding suggests that those interested in owning doodles would benefit from having more information about doodles’ grooming requirements so they can better meet their dog’s welfare needs. Nevertheless, doodle owners reported being highly satisfied with their dogs.

**Abstract:**

“Designer dogs”, which are the hybrid offspring that result from intentionally breeding dogs belonging to different breeds, are an extremely popular pet choice in the United States. Poodle mixes, often called “doodles”, are a very common type of designer dog. However, there are many misconceptions surrounding them, and the reality of owning one may not match the owner’s expectations. For instance, many people believe these dogs to be non-shedding and hypoallergenic, although this is not always the case. This study explored whether the reality of owning a doodle matches owner expectations. For comparison purposes, we also asked owners of non-doodle dogs about their expectations versus reality. Our survey-based study included 2191 owners of doodles and non-doodle dogs recruited via groups of dog owners on Facebook and Reddit. The data showed that, when selecting their dogs, doodle owners were more influenced than non-doodle owners by their dog’s appearance and by the perception that doodles are good with children and are generally healthy. Doodle owners reported being highly satisfied with their dogs; nevertheless, more than twice as many doodle owners than owners of the other groups of dogs reported that their dog’s maintenance requirements, such as their need for regular grooming, were more intensive than they had expected. This finding suggests that those interested in owning doodles would benefit from having more information about their dog’s grooming needs so they can decide whether they have the time and money required to meet their dog’s welfare needs.

## 1. Introduction

In the modern world, dog ownership is a global phenomenon; in the United States alone, approximately 38% of all households own at least one dog, with a population of 77 million dogs nationwide [1]. A recent fad in pet ownership has resulted in the rise in popularity of the “designer dog”, or the hybrid offspring that result from intentionally breeding dogs belonging to different breeds. Examples of designer dogs include poodle hybrids, such as goldendoodles and cockapoos. It is important to note that since designer dogs are a mix of breeds, they are not actual breeds. Thus, the American Kennel Club does not recognize any hybrid dogs in their registry. In response to this fact, the American Canine Hybrid Club was created, according to Urbanik and Johnston (2017), to validate the purchase of these types of dogs for their owners [2].

The trend of crossing poodles with other breeds began in 1989 when Wally Conron, who worked for the Royal Guide Dog Association of Australia, crossbred a Labrador and a poodle in response to the demand for a Labrador’s disposition with a poodle-type coat due to anecdotal reports of poodles being hypoallergenic [3]. Conron confessed that he created the name “labradoodle” because there was little interest in a mixed-breed dog, and he has since expressed regret that it caused a trend of mixing many different breeds with poodles [2]. He has lamented that because of his creation, poodle crosses are bred in puppy mills and through other irresponsible breeding practices [4]. Around the time that Wally Conron was creating the labradoodle in Australia, a man named Wallace Havens began breeding puggles (pug/beagle crosses) in the United States, and they have also become very popular [2].

Designer dogs have become particularly popular in recent years. For example, Nationwide Pet Insurance has reported that from 2013 to 2021, the popularity of poodle crosses among Nationwide policies increased by 160.3% [5]. The popularity of these dogs may be due in part to the many beliefs surrounding them. For example, people commonly believe that these types of dogs are healthier and have fewer inherited health defects than their parent breeds [2]. Indeed, Nationwide Pet Insurance recently reported that owners of poodle crosses are significantly less likely to have submitted a claim for cancer diagnosis and treatment than are owners of their purebred counterparts [5].

Nevertheless, the reality is that being crossbred does not inherently make a dog healthier. Even though crossing two breeds can indeed increase heterozygosity and reduce disease-causing alleles in a breed [6], it is key to choose lineages carefully and avoid breeds with many inherited genetic disorders. For instance, the common cross of a Labrador and a standard poodle can have a detrimental impact on the health and welfare of the offspring since both breeds are susceptible to similar genetic disorders, such as hip dysplasia and diseases of the eyes and joints [6]. Furthermore, another study found no difference between purebred and crossbred dogs in the prevalence of 13 diseases [7]. Thus, the idea that crossbred dogs are healthier and impervious to genetic disease is not based on fact, as they can inherit the same conditions that purebred dogs can [7].

There is another common misconception that designer dogs have better temperaments than their parent breeds. However, the temperaments of designer breeds often fall somewhere between those of their two parent breeds [8]. In a survey completed by 5141 dog owners, labradoodles were not found to display any significant differences in fourteen behavioral categories compared to standard poodles or Labradors [8]. One exception to this is the goldendoodle, which scored significantly higher than their parent breeds in some problematic behaviors, including dog-directed aggression, dog-directed fear, and stranger-directed fear [8]. Therefore, while people may acquire crossbred dogs thinking that they will have better health and temperaments than their parents, those assumptions are not necessarily true.

Other common misconceptions about designer dogs include that they are non-shedding, hypoallergenic, and good family dogs [9]. Though this is certainly true for some individual designer dogs, it is impossible to generalize about a type of designer dog as a whole. In fact, there is considerable variability even among puppies from the same litter; for instance, an estimated one-third of labradoodles within the same litter end up having a Labrador-type coat, meaning that they shed and can induce allergies [10]. Furthermore, a study that compared the allergen levels in hair and coat samples from various dog breeds concluded that there is no evidence to classify certain dog breeds as hypoallergenic [3]. Additionally, dogs such as labradoodles also require extensive at-home and professional grooming, and this reality may have serious ramifications for dog welfare if regular grooming is neglected [11]. This is important information for prospective dog owners to be aware of, as it can better prepare them for what to expect and thus enable them to meet their dog’s welfare needs.

While designer dogs are extremely popular, they are also controversial. A common issue that people raise with these types of dogs is that they exist simply to create demand for a new “product”. There is controversy over breeding dogs simply to make them more visually appealing to people and exploiting them, charging thousands of dollars per puppy [12]. Breeders of both purebreds and hybridized designer dogs commonly sell puppies for thousands of dollars. Despite the high cost at which these dogs are sold, dog breeding in the United States remains unregulated, and not all breeders carefully consider the health or behavior of the parents or offspring [2]. Selling dogs without regard for their health or behavior represents a type of commodification in which people use the dogs as objects simply to make money and exemplifies the controversial issue of treating animals as objects or status symbols rather than as subjects in their own right. Though many people acquire dogs for intrinsic reasons and view them as individuals, others may acquire their dogs due to extrinsic motivations, such as seeking acknowledgement from others or wanting status or control [12]. Such motivations can result in prioritizing the dog’s appearance over their health and behavior [13].

Similarly, a notable ethical implication related to the rising popularity of designer dogs is the correlation between the demand for these types of dogs and unethical breeding practices. Although the American Kennel Club recognizes 199 pure breeds [14], an estimated 20 percent of dogs bred in puppy mills are designer dogs [10]; thus, their popularity likely fuels commercial breeding practices that prioritize profit over animal welfare. This causes welfare issues not only for the puppies themselves but also for the parent dogs that live and breed in those environments [15]. Dogs purchased as puppies from these establishments often display greater rates of fear, house soiling, separation-related problems, and aggression towards people and other dogs than do puppies from other breeding establishments [16]. Additionally, dogs formerly used as breeding stock in puppy mills have higher rates of general fear, health problems, and house soiling compared to other pet dogs [17].

Like designer dogs, brachycephalic dogs are extremely popular, frequently victims of inhumane breeding practices, and bred for their appearance rather than their health or behavior [18]. These similarities can provide insight into the practices of designer dog acquisition. For instance, people acquire brachycephalic dogs mostly due to their appearance, followed by their size, and the perception that they are good with children and a good companion breed. Health is not a key factor influencing decisions to acquire them, even though they often have serious chronic health conditions [19]. Owners of these dogs are also more likely to buy them from a puppy-selling website and are less likely to ask about their health and meet their parents than are owners of non-brachycephalic dog breeds [19].

People acquire dogs with certain expectations about what dog ownership will be like. The expectations and realities of dog owners are very important to consider since a mismatch between them can result in people relinquishing their dogs to shelters and rescue groups [20]. A factor that may affect the expectations of dog owners is their previous experience owning dogs. People who have had a dog before may have more realistic expectations and thus do not have to adjust their expectations after acquiring a dog. First-time dog owners, on the other hand, often must change their expectations shortly after acquisition [21]. Previous experience with a specific breed can also greatly impact people’s decisions to acquire a dog, with many people attributing prior experience with dogs to being highly influential in their breed acquisition decisions [22]. Interestingly, despite the chronic health conditions associated with brachycephalic breeds, owners of these breeds are highly likely to report intentions of reacquiring the same breed [23]. The likelihood of reacquisition is also increased for owners who report having a strong relationship with their dog and for first-time dog owners. On the other hand, owners of dogs that experience numerous health problems and worse behaviors than expected are less likely to acquire the same breed in the future [23].

While a fair amount is known about the expectations and realities of dog ownership in general, there is an information gap regarding designer dog ownership. Designer dog owners may have unique or very specific expectations. Interviews of dog owners suggest that designer dog owners expect their dogs to have stable temperaments, be healthier than other dogs, and have hypoallergenic, non-shedding coats [9]. Power (2012) also reported that these assumptions indicate an overall perception of them fitting into domestic life better than other dogs [9]. If such expectations do not match reality, this may create tension in the human-dog relationship and even result in the dog’s relinquishment [24]. Additionally, these misconceptions have ethical implications since they may fuel unethical breeding and, thus, commodification and compromised welfare of many dogs. This study addressed this information gap by exploring the expectations versus reality of designer dog owners compared to non-designer dog owners. Specifically, we compared responses from owners of poodle crosses with those from owners of purebred and mixed-breed dogs. Due to the many misconceptions surrounding designer dogs, we predicted that there would be a greater mismatch between the expectations and reality of poodle-cross owners compared to those of other dog owners.

## 2. Materials and Methods

### 2.1. Participants

Study participants included owners of purebred dogs, mixed-breed dogs, and poodle crosses. Poodle crosses had to be the offspring resulting from the intentional mix of at least two purebreds, one of which was either a mini, toy, or standard poodle. We included any generation of poodle mix, not just the first generation offspring, and advertised the survey in several Facebook groups and the r/dogs group on Reddit. The study sample consisted of 2191 total dog owners: 689 owners of poodle crosses (“doodles”), 854 owners of other mixed breeds, and 648 owners of purebred dogs.

### 2.2. Measures and Procedure

Upon clicking the survey link, participants first completed the consent form and then shared the name of their dog. We instructed people with multiple dogs to complete the survey for the dog they had owned the longest. We then referred to that dog by name throughout the survey so that participants remained focused on the correct dog.

The first section of the survey included questions created and validated by Packer et al. (2019) [18]. Specifically, it consisted of four questions targeting owners’ expectations and reality regarding dog ownership. The questions addressed veterinary costs, exercise requirements, maintenance levels, and overall behaviors of the dog. The veterinary costs and exercise levels had responses of “less than expected”, “met expectations”, and “more than expected”; the behavior and maintenance questions had responses of “worse than expected”, “met expectations”, and “better than expected”.

The second part of the survey included nine questions that comprise the perceived costs subscale of the validated Monash Dog Owner Relationship Scale (MDORS). These questions addressed whether people viewed caring for their dog as a chore and how much of a burden it was to care for them. Each item had a 5-point ordinal scale rating with 0 being “never” or “completely disagree”, and 4 being “very often” or “completely agree” (α = 0.84) [25].

The third section of the survey came from Bouma et al. (2020a) [22], which described a brief measure that was inspired by Hsu and Serpell’s (2003) validated Canine Behavioral Assessment and Research Questionnaire (C-BARQ) [26]. This section asked 16 questions about canine behaviors related to the dog’s excitability, aggression, fear and anxiety, training and obedience, and miscellaneous problem behaviors. Participants answered these questions and indicated which behaviors their dog exhibited. All questions were on a 5-point ordinal rating scale, with 0 being “absent/never” and 4 meaning “very often”.

The fourth section of the survey came from Bouma et al. (2020) [27]. These seven questions addressed owner satisfaction with their decision to acquire their dog. Participants answered these questions using a 5-point scale, with answer choices ranging from “completely disagree” to “completely agree” (α = 0.77) [27]. We reverse-coded four of the questions and calculated the mean satisfaction scores for each participant.

The next section of the survey asked people to indicate the degree to which each of 15 potential factors influenced their decision to acquire their specific type of dog. The response options came from Packer et al. (2017) and included factors such as dog appearance, temperament, and popularity. Participants scored each factor on a 5-point scale from “not influential at all” to “extremely influential” [19].

The last section of the survey collected basic demographic information about the participant and their dog. This section of the survey included several categorical questions. For instance, for the first question, “What type of dog do you have?” participants had the option of selecting “purebred dog”, “hybrid dog”, or “mixed breed not typically considered a hybrid”. The next question included an open-ended text box so participants could provide more details about what type of dog they had. The questions “Is this the first dog that you have owned?”, “Have you owned this type of dog before?”, and “Did you speak to any professionals about your breed choice before acquisition?” were categorical yes/no questions. Two multiple-choice questions asked participants to specify from where they acquired their dog and to whom they spoke before acquisition. We asked the following to people who indicated that they had purchased their dog from a breeder: “Did you meet at least one of [dog name]’s parents?” We asked this in an attempt to determine whether they purchased their dog from a puppy mill, as these types of facilities are less likely to allow people to meet the parents [28]. The final question was optional and gave participants the opportunity to share three ways their dog met their expectations and three ways that they did not meet those expectations. A copy of the full survey has been included in the manuscript’s Appendix A.

### 2.3. Data Analysis

When analyzing the first four questions regarding expectations versus reality, we scored “worse than expected” as a 0 and “met expectations” and “better than expected” together as a 1. We examined dog type (i.e., doodle, purebred, or mixed-breed) in relation to answers to each of the four questions individually with Chi-square analyses since the questions were not necessarily related to each other. For these Chi-square analyses and the analyses described below, rather than using the traditional α value of 0.05, we opted to use a more conservative α value of 0.01 due to the many tests run. When results of the 3 × 2 Chi-square tests were significant, we ran pairwise, post-hoc comparisons using a Chi square test of association with *p*-values adjusted using the False Discovery Rate method.

We ran general linear models to examine whether dog type, owner age category, whether children were in the household, and whether this was the owner’s first dog predicted each of the following: MDORS scores, behavior scores, and owner satisfaction scores. These scores were calculated by averaging the responses that comprised each measure. The MDORS and behavior scores were right skewed, and so we log-transformed the scores (e.g., log(MDORS scores)) prior to running the general linear models. The satisfaction score was left skewed, and so we used the following formula to log-transform those scores: −log10 (max(satisfaction score + 1) − satisfaction score). The transformed scores were then normalized using the “scale” function in R, and we used the traditional α value of 0.05 for the general linear models.

In the section in which participants rated how influential 15 different factors were, we assigned a one or zero to each score (a 1 for “Extremely Influential” and “Very Influential”, and a 0 for everything else) and then ran a Chi square test for each factor comparing across the three dog groups. Lastly, to analyze answers to the free-response questions, we conducted a qualitative content analysis. As part of the content analysis, we carefully read through the responses and then created categories into which the answers fell based upon the trends that we observed. Then we read through the responses a second time. This time, we assigned each response to one of the categories we had created. We then tallied up the number of instances of each category for each of the three dog breed groups.

## 3. Results

Of the 2987 individuals who started the survey, 2191 completed it, yielding a 73.3% completion rate. We analyzed completed responses, which included 689 owners of doodles, 854 owners of other mixed breeds, and 648 owners of purebred dogs. Eighty-two percent of participants identified as female; 15.7% identified as male; 1.3% identified as non-binary/third gender; and 0.6% opted not to share their gender. Forty-one percent of participants were in the 18–34-year-old age category; 38.4% were in the 35–54-year-old category; and 20.6% were 55+ years old. Ninety percent of participants were white, with the other race categories comprising less than 5 percent of the sample. In terms of ethnicity, 6.0% of participants identified as Hispanic.

We compared the characteristics of owners of doodles, purebreds, and mixed breeds, and these results are listed below in Table 1. One notable difference between the three groups is that nearly twice as many (41.8%) doodle owners had children in the home, compared to 21.8% of mixed-breed owners and 22.8% of purebred owners.

Participants acquired their dogs from a variety of sources (Table 2). The majority of doodle and purebred dog owners acquired their dogs from breeders (82.9% and 60.7%, respectively), whereas 75.5% of mixed-breed owners adopted their dogs from a shelter or rescue. Sixty-seven percent of purebred owners who purchased their dogs from a breeder met at least one parent, compared to 57.1% of doodle owners and 55.3% of mixed-breed owners who purchased their dogs from a breeder.

Owners of the three dog types indicated that different factors influenced their decision to acquire their dog. The factors that differed significantly across groups, along with the percentage of dog owners within each group who listed the factors as influential, are reported in Table 3. For instance, the appearance of the type of dog was much more influential for doodle owners than for the other two groups (χ^2^ = 65.16, df = 2, *p* < 0.001). Fifty percent of doodle owners were influenced by appearance when acquiring their dog, compared to 36.7% of purebred owners and 30.0% of mixed-breed owners (post-hoc comparisons: purebreds vs. doodles: *p* < 0.001; mixed-breeds vs. doodles: *p* < 0.001; mixed-breed vs. purebred: *p* = 0.006).

The assumption that a particular type of dog was good with children factored differently into different types of dog owners’ decisions. The perception of a particular type of dog being good with children was influential for 53.7% of doodle owners, compared to 27.2% of purebred owners and 8.8% of mixed-breed owners (χ^2^ = 379.48, df = 2, *p* < 0.001; post-hoc comparisons: doodle vs. mixed-breed: *p* < 0.001; doodle vs. purebred: *p* < 0.001; mixed vs. purebred: *p* < 0.001). Given these significant differences, we also compared the influence of being good with children between doodles and their parent breeds, or breeds of dogs that are commonly mixed with poodles (i.e., Golden Retrievers, Labradors, and cocker spaniels). We found that, compared to the 53.7% of doodle owners who reported the dog being good with children as influential in their acquisition decision, 37.9% of owners of parent breeds reported this factor as influential (χ^2^ = 19.40, df = 1, *p* < 0.001).

An important component of day-to-day life with a dog is the dog’s behavior. The median behavior score of all three groups of dogs was 1.69, suggesting that most dog owners experienced few behavior problems. The general linear model showed that there was no significant difference in behavior across the three dog types (Table 4). However, people in the older two age groups were also more likely to assign their dogs lower behavior scores, which were indicative of fewer problem behaviors, than were participants in the younger age group. Finally, those with children were more likely to assign their dogs higher behavior scores than those without children; similarly, people for whom this was their first dog were more likely to assign their dog higher behavior scores.

The median perceived costs score from the MDORS was 1.89 for mixed-breed dogs and 1.78 for dogs in the other two groups. The general linear model examining associations between dog type, owner age, whether there were children in the home, and whether the owner was a first-time dog owner and MDORS perceived costs scores indicated no significant difference in perceived costs between the three dog types (Table 5). However, owners in the youngest age category rated the perceived costs higher than those in the oldest age category. Likewise, those with children were more likely to score the perceived costs higher than those without children, and those for whom this was their first dog were more likely to score the perceived costs higher than people who had owned a dog previously.

Expectations versus reality regarding maintenance levels such as grooming differed greatly across the three groups. Twenty-four percent of doodle owners indicated that their dog’s maintenance levels were worse than they expected, compared to only 11.0% of purebred owners and 10.1% of mixed-breed owners (χ^2^ = 67.55, df = 2, *p* < 0.001; post-hoc comparisons: doodle vs. purebred: *p* < 0.001; doodle vs. mixed-breed: *p* < 0.001; mixed-breed vs. purebred: *p* > 0.01).

One of the main variables that we examined was owner satisfaction scores. The median satisfaction scores were similar across groups: doodle owners had a median satisfaction score of 4.57; mixed-breed owners had a median satisfaction score of 4.29; and purebred owners had a median satisfaction score of 4.43. The general linear model showed that satisfaction scores for doodle owners were higher than for those with purebreds or mixed breeds (Table 6). Additionally, satisfaction scores were higher for people in the older two age groups than in the youngest age group. Those with children had lower satisfaction scores than those without, and those for whom this was their first dog also had lower satisfaction scores than those who had had a dog previously.

Qualitative analysis of the last survey question, which asked about three ways the dog met expectations and three ways the dog did not meet expectations, suggests doodle owners’ expectations prior to acquiring their dog did not match reality. Of the 689 total doodle owners, 639 of them answered the question addressing ways their dog met their expectations, and 479 answered the question asking about ways their dog did not meet their expectations. Of the doodle owners who answered the free-response question, 87 (18.2%) doodle owners lamented about the cost and frequency of grooming, and 32 (6.7%) complained specifically about how much daily brushing was required, stating that they expected their dog to have a low-maintenance coat and instead ended up with the opposite. Comparatively, 20 (3.1%) owners of mixed-breed dogs complained of their dogs’ grooming needs, and 9 (1.4%) lamented their dogs’ general maintenance. Of the purebred owners who answered the question, 25 (5.2%) complained of their dogs’ grooming needs, and 21 (4.4%) complained of their dogs’ general maintenance. An owner of a Bernese Mountain dog/poodle mix (often called a “Bernedoodle”) responded that the grooming expense was “nearly double [the] expected cost”. A goldendoodle owner reported, “The upkeep on grooming is a lot more than expected”. And, finally, another person echoed the responses of numerous other participants when they said, “I’m tired of grooming [my goldendoodle]”.

Complaints about grooming were second only to negative statements about barking. Of the 479 doodle owners who answered the free-response question about how their dog did not meet their expectations, 122 (25.5%) stated that their dog barked more than expected. Comments such as “Barks more than I expected”, “Excessive barking”, and “Uncontrolled barking” were very common. One participant said that their goldendoodle was “not as quiet as I had hoped—she loves to bark”. In comparison, out of the 637 mixed-breed owners and 479 purebred owners who answered the question, 118 mixed-breed owners (18.5%) and 81 purebred owners (16.9%) listed barking as an expectation not met.

A qualitative analysis of the free-response question also revealed other miscellaneous expectations not met for doodle owners. The owner of a sheepdog/poodle hybrid (often called a “sheepadoodle”) wrote, “It’s a shedding dog and [I] was told it wouldn’t be”. Similarly, an owner of a lab/poodle cross (“labradoodle”) indicated that her dog sheds, and she “didn’t realize there were labradoodle generations that were less hypoallergenic”. Sixty-four (13.4%) doodle owners complained about their dogs having a range of health issues, including allergies, stomach issues, and orthopedic problems.

Free-response answers also showed expectations not being met due to an apparent misunderstanding of genetics. For instance, one person reported that her Bernese Mountain dog/poodle cross was “not quite as obedient as I understood this breed to be”, though of course it is a mix of breeds and thus there is no breed standard. Fifty-five (11.3%) doodle owners complained about their dog’s appearance, noting changes in their dog’s coat type, apparently misunderstanding that, due to puppies being mixes of more than one breed, it is difficult for breeders to predict what the dog will look like as an adult.

It is possible that the misconceptions about doodles come in part from the breeders themselves. In the free-response question, some people reported being misled by the breeder. For instance, one participant said that “grooming was not truthfully explained” by the breeder. Another person said that her dog was “larger in size than I was advised”; yet another person reported that her dog was “undersocialized by [the] breeder”. One participant complained that her expectations were not met regarding “the breeder’s commitment”. Finally, the owner of a sheepdog/poodle mix said that their dog was “supposed to be a Bernedoodle” (a Bernese Mountain dog/poodle mix).

Though most doodle owners reported that their dogs failed to meet expectations in one way or another, there were still many ways that their dogs did meet expectations. 284 (41.0%) doodle owners noted that their dog’s temperament met their expectations. In the free-response section, many people described their doodles as being “affectionate”, “cuddly”, “loving”, and “friendly”. Additionally, 250 (36.1%) doodle owners reported viewing their dogs as particularly intelligent or smart. Finally, 198 (31.0%) doodle owners described their dogs as being good companions, offering good company, and being members of the family.

## 4. Discussion

Findings from this study indicate that owners of doodle dogs have certain expectations of what it will be like to own these dogs. These expectations are not always rooted in reality, however. Appearance is significantly more important to these owners than it is to owners of mixed-breed and purebred dogs, and appearance strongly influenced their choice to acquire these dogs. Doodle owners were also disproportionately influenced by the preconceived notion that doodles are good with kids and that they are generally a healthy breed, which is not consistently the case.

Overall, our results indicate that people acquiring doodles may make their acquisition decisions based on both aesthetics and misconceptions about the dogs, thinking them to be healthier and better with kids than other dog breeds. These results support previous findings by Power (2012), who observed similar misconceptions [9]. Our findings also echo those of Sandøe et al. (2017), who observed that dog owners may also acquire specific breeds for the attention that their appearance generates, thus prioritizing the dog’s appearance over their health and behavior [13]. The fact that many doodle owners made acquisition decisions based on the appearance of the dog indicates that they may have been prioritizing aesthetics over health and behavior. Regardless of the reason, these preconceived notions and misconceptions about doodles have evidently led to some degree of mismatch between expectations and reality of doodle dog ownership.

A significant way that expectations of owning a doodle did not meet reality was regarding the dogs’ maintenance requirements, particularly grooming. Compared to owners of purebred and mixed-breed dogs, more than twice as many doodle owners reported that their dog’s maintenance was worse than expected. Doodle owners also reported this in the free-response questions, commenting that the maintenance of their dog’s coat was not only much more time-intensive than expected but also much more expensive.

While owner satisfaction was highest among doodle owners, there was no difference between owners of different dog types in terms of perceived costs of dog ownership or problematic behaviors. It seems that more influential to perceived costs of dog ownership is the presence of children and first-time dog ownership. Those with children and those for whom this was their first dog reported a higher caregiver burden than those without children and those who had owned dogs previously. That is, they were more likely to report that caring for their dog was challenging, a chore, and costly in terms of money and time. Bouma et al. (2020a) reported similar findings, observing that first-time dog owners often must change their expectations shortly after acquiring their dog [21].

Given that over 40% of doodle owners who reported acquiring their dog from a breeder did not meet either of their dog’s parents, a large proportion of doodle owners may have purchased their dogs from puppy mills. This is unsurprising, as Packer et al. (2017) found a similar trend in owners of brachycephalic breeds, which are experiencing a level of popularity comparable to that of doodles [19]. A characteristic of responsible breeders is that they commonly allow people to meet the puppy’s parents. Puppy mills, meanwhile, typically do not allow people to meet parents and may even meet prospective dog owners in a neutral location so that their clients do not visit the facility itself [29]. The difference observed regarding purebred and doodle owners in this respect suggests that doodle owners may not be as well informed about best practices regarding pet acquisition as are many purebred owners.

Answers to the free-response question indicated that misconceptions and misunderstandings may stem from the breeders themselves. Some doodle owners reported that their breeders misled them or did not truthfully explain their dog’s grooming requirements.

Regardless of the source of owners’ misconceptions, future dog owners would benefit from having access to more information about doodles in particular and dog ownership more generally. Ensuring prospective dog owners have access to such information could have important implications not only for the welfare and satisfaction of dog owners but also for the dogs themselves. Dog owners who have sufficient information when acquiring a dog can make a properly informed decision. For example, when dog owners have a better understanding of their dog’s grooming requirements, they are better equipped to meet their dog’s welfare needs.

### Limitations

One of the limitations of our study is that the term “doodle” refers to a category of dogs rather than one particular breed. The label spans many types of dogs that are the result of crossbreeding countless breeds with poodles. Thus, there was great variability among participants and their responses. Furthermore, we relied on owners’ reports about what type of dog they had rather than on genetic screening, which could determine dog type more objectively. It is possible that genetic testing would have shown that some owner-reported doodles were more closely aligned with our definition of a mixed-breed and vice versa. Another weakness of our study is that we were unable to estimate how many doodles live in the United States. Because poodle crosses are not purebreds, there is no official registry such as the American Kennel Club to track the size of the doodle population.

One possible source of bias is that the survey does not include responses from owners of dogs that have been relinquished or rehomed, possibly due to the dog not meeting owner expectations. Though we did not capture responses from such individuals in our study, further research on this topic would help address the extent to which mismatches between doodle owners’ expectations and reality impact doodle welfare.

Another potential source of bias in our study design came from using recruitment of participants through Facebook and Reddit groups. This presented a limitation as it only targeted dog owners who used Facebook and Reddit and were inclined to join these groups. This may include an audience that is generally younger and that may be particularly enthusiastic about and satisfied with their dogs. However, as a large portion of the population has Facebook accounts (71.43% of the United States population as of 2022) [30], we believe that this still captured a wide audience and may have even resulted in more participants than alternative sampling methods would have.

## 5. Conclusions

This study addressed an important disconnect between doodle owner expectations versus reality. Doodle owners’ decisions to acquire their dogs were disproportionately influenced by the dog’s appearance and by the perception that doodles are good with children and are generally healthy breeds. Doodle owners commonly reported that the maintenance requirements associated with their dogs were more intensive than they had expected. Indeed, more than twice as many doodle owners than owners of purebred and mixed-breed dogs reported that maintenance levels, including grooming requirements, were worse than they expected. Furthermore, open-ended responses suggest that some breeders are misleading prospective dog owners and do not honestly explain what to expect from their dogs in terms of grooming needs and potential coat type. Nevertheless, doodle owners were highly satisfied with their dogs.

## Figures and Tables

**Table 1 animals-12-03247-t001:** Demographic characteristics of owners of each dog breed group.

Type of Dog	18–34 Years	35–54 Years	55+ Years	Children in Home	First-Time Dog Owner
Doodle	25.8%	45.1%	29.0%	41.8%	25.7%
Mixed-breed	51.5%	34.4%	14.1%	21.8%	34.4%
Purebred	43.4%	36.4%	20.2%	22.8%	25.5%

**Table 2 animals-12-03247-t002:** Sources of each type of dog, including the percentage of people who bought their dog from a breeder and met the parents.

Type of Dog	Shelter or Rescue	Pet Store	Friend/Family Member	Other	Breeder	Met Parents
Doodle	3.2%	3.5%	5.2%	5.2%	82.9%	57.1%
Mixed-breed	75.5%	0.8%	8.0%	11.2%	4.5%	55.3%
Purebred	19.3%	3.1%	8.6%	8.3%	60.7%	67.1%

**Table 3 animals-12-03247-t003:** Factors that influenced people to acquire their dog type, with percentages of each group of dog owners that listed each factor as influential. ◊ denotes that each group differed significantly from the others; ○ denotes significant differences between mixed breeds and doodles and between mixed breeds and purebreds; and □ denotes significant differences between purebreds and doodles and between purebreds and mixed breeds.

Factor	Doodles	Purebred	Mixed-Breed
Appearance ◊	49.9%	36.7%	30.0%
Popularity ◊	22.4%	15.4%	4.0%
Good with kids ◊	53.7%	27.2%	8.8%
Good companion breed ◊	83.2%	67.4%	35.1%
Working ability ◊	10.0%	18.5%	4.0%
Generally healthy breed ◊	51.7%	43.7%	25.5%
Long life expectancy ○	37.7%	33.5%	21.7%
Breed suited to lifestyle ○	69.8%	70.0%	43.7%
Breed easy to care for □	29.0%	36.6%	27.2%
Recommended by friend/family member ○	22.5%	18.2%	8.1%

**Table 4 animals-12-03247-t004:** The beta estimate and standard error for each factor in the general linear model examining the relationship between breed type, owner age, the presence of children in the home, whether this was the owner’s first dog, and owner-reported dog behavior.

Variables	Estimate	Std. Error
Intercept	0.06	0.06
Mixed-breed (ref: Doodle)	−0.02	0.05
Purebred (ref: Doodle)	−0.04	0.05
Owner aged 35–54 (ref: Owner aged 18–34)	−0.12 *	0.05
Owner aged 55+ (ref: Owner aged 18–34)	−0.41 ***	0.06
Children in home (ref: No children)	0.16 **	0.05
First dog (ref: Not first dog)	0.17 ***	0.05

* *p* < 0.05, ** *p* < 0.01, *** *p* < 0.001.

**Table 5 animals-12-03247-t005:** The beta estimate and standard error for each factor in the general linear model examining the relationship between breed type, owner age, the presence of children in the home, whether this was the owner’s first dog, and owner perception of perceived costs of ownership.

Variables	Estimate	Std. Error
Intercept	0.27	0.01
Mixed-breed (ref: Doodle)	0.00	0.01
Purebred (ref: Doodle)	0.00	0.01
Owner aged 35–54 (ref: Owner aged 18–34)	−0.01	0.01
Owner aged 55+ (ref: Owner aged 18–34)	−0.03 ***	0.01
Children in home (ref: No children)	0.02 ***	0.01
First dog (ref: Not first dog)	0.03 ***	0.01

*** *p* < 0.001.

**Table 6 animals-12-03247-t006:** The beta estimate and standard error for each factor in the general linear model examining the relationship between breed type, owner age, the presence of children in the home, whether this was the owner’s first dog, and owner satisfaction.

Variables	Estimate	Std. Error
Intercept	0.20 ***	0.06
Mixed-breed (ref: Doodle)	−0.25 ***	0.05
Purebred (ref: Doodle)	−0.21 ***	0.05
Owner aged 35–54 (ref: Owner aged 18–34)	0.19 ***	0.05
Owner aged 55+ (ref: Owner aged 18–34)	0.28 ***	0.06
Children in home (ref: No children)	−0.14 ***	0.05
First dog (ref: Not first dog)	−0.47 **	0.05

** *p* < 0.01, *** *p* < 0.001.

## Data Availability

The data presented in this study are available on request from the authors.

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
