# Peer review of "Expectations versus Reality of Designer Dog Ownership in the United States"

_animals, 2022, doi:10.3390/ani12233247_

Round 1

Reviewer 1 Report

This is an interesting topic and I am pleased the authors are investigating it.

My comments are only minor and are made solely with the aim of improving the manuscript.

Line 106. There are responsible and irresponsible breeders. I am not sure about the use of “commonly”. I know many responsible purebred breeders who do consider the long term welfare of the breed and test for a variety of genetic conditions. The problem is lack of regulation and accountability.

Line 106. This reference [2] is for the whole book. Were the opinions attributed to this reference scattered throughout the book or limited to a chapter? If so, can the reference reflect this please? I would like to view the original research.  

Line 145. I found this a little confusing. Brachycephalics often have health problems but this sentence suggests that owners of brachycephalics are likely to acquire another.

Measures and Procedures

Is it possible to have a copy of the survey in additional information so the wording of questions and answer options is available?

Lines 208 & 248. Line 208 states that options included ‘extremely influential’ while Line 248 uses the terminology ‘very important’. Again, providing the survey would be appreciated.

Line 273, Lines 429-439 & Table 2. I would question the results to the question regarding meeting a parent. According to Table 2, 75% mixed breed dogs were acquired from shelters or rescue groups and 55% owners of mixed breed dogs met the parents. This seems unlikely. How many participants answered this question?

Tables 4,5 & 6. Please define *, **, and ***

Lines 359-364. It would be interesting to know how many owners of purebreds and mixed breeds made comments about barking for comparison.

Line 442. I believe it should be “grooming” rather than “breeding”.

Lines 459-466. I believe there is another potential source of bias that should be acknowledged. This survey does not include those who rehomed or relinquished their dogs, perhaps because they did not meet expectations. Did you discover the proportion of “doodles” in shelters or rescue groups?

Author Response

For responses to comments, please see attachment. 

Reviewer 2 Report

Thank you for an interesting paper.

The following comments are intended to help to improve the quality of the paper and to be constructive.

1.      When outlining the surveys from which you took questions for your survey, it would be helpful to indicate which of these are validated surveys, and which are not. For example, the C-BARQ is validated but I am not sure about the others. For validated studies, what is the effect of using a subset of their questions on the validity of the study?

2.      By including not just first generation offspring in the group ‘doodles’ – is there a danger that some dogs were included who might have been better classified as ‘mixed breed’ dogs, and how might this have affected the results?

3.      Please clarify the approach used for qualitative analysis. It looks like you have used content analysis, but please expand on how this was performed and checked.

And finally, some minor points/typos

a.      Please clarify the spelling of Bouma p4, lines 193 and 200

b.      P6, lines 261-263, the reporting is rather clumsy. If 6% identified as Hispanic, then that is not less than 5%, which is how it currently reads. Perhaps reordering (….identified as white, 6% as Hispanic and less than 5% as all other races)

c.      Tables 4-6, I assume that ‘owner ago’ should be ‘owner age’?

d.      P10 line 425 – I think it would be good to explain ‘caregiver burden’ as some readers may not recognise this term

e.      P10 line 442 should ‘breeding’ be ‘grooming’?

f.       P11 line 471 – please replace ‘kids’ with ‘children’ to avoid colloquialism

Author Response

(The authors gave the same response as above.)
